# OpenReview forum: "Benchmarking Intent Awareness in Prompt Injection Guardrail Models"
_ICLR.cc/2025/Workshop/BuildingTrust — Submitted to BuildingTrust_

### Official Review · Reviewer_gtpA · 2025-02-25
**Novel Pipeline and Datasets to Address Benign Prompt Injection Identifications; Lacking in Clarity, Organization, and Clear & Wide Usability/Significance**

**Rating:** 5
**Confidence:** 3

**Review:**

This workshop paper highlights a key issue in prompt injection LLM defense: benign examples can lead to high FPR, and on intent-aware datasets, can exhibit both high FPR and FNR. Perhaps, in non-intent-aware settings, this issue is more or less alleviated. The authors propose a new datasets that can help correct and analyze FP/FN behaviors in LLMs, introduce a new framework for generating challenging  tough prompts, and introduce a prompt injection defense LLM with original and introduced datasets that performs better on benchmarks. Overall, I find this paper to be very hard to parse due to the formatting of sentences and paragraphs (ie using an acronym before it is introduced), walls of text that introduce a bunch of variables/notation, and lack of clarity in description. It appears this paper highlights an issue and addresses it with the datasets and model, but it is not clear how much improvement this model brings as it only has results for one dataset (PPC). Intent-Inject IRS Analysis is in a paragraph rather than incorporated into the table, which is also hard to read (as described about clarity earlier). In summary:

Pros:

- Table 1 gives good intuition on the difference between I-I and N-I-I
- Novel datasets / pipeline can help bridge gap between extreme FPR/FNRs with benign examples, which can potentially happen frequently in real life
- Proposed model performs better across one dataset

Cons:

- Pretty hard to parse paper (notes above) with notations, setup, results, etc.
- It would be nice to have more setup on why this is important IRL - for example, these benign examples can happen very frequently in LLM use cases or education, cannot be so extreme to label them as adversarial injections
- Only show improvement across only 1 dataset

---

### Official Review · Reviewer_KjeJ · 2025-03-02
**The paper presents a framework for evaluating intent-aware prompt injection attacks in Python chatbots and airline booking assistants, showing improved performance with their model, but lacks comprehensive comparisons and broader domain evaluations.**

**Rating:** 4
**Confidence:** 3

**Review:**

The paper provides a systematic framework for evaluating intent-aware prompt injection attacks and conducts experiments with existing guardrails across two domains: Python programming chatbots and airline booking assistants. The authors propose and evaluate their own dataset and model, showing improved performance.

However, the paper could be strengthened by including more examples and illustrations comparing the author’s trained model with other models.

A limitation of the paper is its focus on only two specific application domains (PPC and FBA). It would have been beneficial to use additional domains and safety benchmarks for more robust evaluations.

Additionally, for the comparison in Table 2, significant models such as LlamaGuard, WildGuard, and other existing LLMs are missing. For example, how would the existing small Llama-3.1/2-1/3/8B-Instruct model perform using prompt engineering for the given task compared to the author's trained model?

I believe the paper needs to be strengthened more at this point.

---

### Decision · Program_Chairs · 2025-03-04

Reject